# Electrochemical Impedance as an Assessment Tool for the Investigation of the Physical and Mechanical Properties of Graphene-Based Cementitious Nanocomposites

**DOI:** 10.3390/nano13192652

**Published:** 2023-09-27

**Authors:** Eirini Tziviloglou, Zoi S. Metaxa, George Maistros, Stavros K. Kourkoulis, Dionysios S. Karousos, Evangelos P. Favvas, Nikolaos D. Alexopoulos

**Affiliations:** 1Research Unit of Advanced Materials, Department of Financial and Management Engineering, University of the Aegean, 82132 Chios, Greece; 2Department of Chemistry, International Hellenic University, 65404 Kavala, Greece; 3ADVISE, 17 Gymnasiarchou Madia Str., 82132 Chios, Greece; 4Department of Mechanics, National Technical University of Athens, 15780 Athens, Greece; 5Institute of Nanoscience and Nanotechnology, N.C.S.R. “Demokritos”, Patr. Gregoriou E & 27 Neapoleos Str., 15341 Agia Paraskevi, Greece

**Keywords:** electrochemical impedance spectroscopy, graphene-based cementitious composites, graphene nanoplatelets, electrical resistivity, porosity, fracture toughness, non-destructive evaluation

## Abstract

This investigation explores the potential of electrochemical impedance spectroscopy (EIS) in evaluating graphene-based cementitious nanocomposites, focusing on their physical and structural properties, i.e., electrical resistivity, porosity, and fracture toughness. EIS was employed to study cement mixtures with varying graphene nanoplatelet (xGnP) concentrations (0.05–0.40% per dry cement weight), whereas flexural tests assessed fracture toughness and porosimetry analyses investigated the structural characteristics. The research demonstrated that the electrical resistivity initially decreased with increasing xGnP content, leveling off at higher concentrations. The inclusion of xGnPs correlated with an increase in the total porosity of the cement mixtures, which was indicated by both EIS and porosimetry measurements. Finally, a linear correlation emerged between fracture toughness and electrical resistivity, contributing also to underscore the use of EIS as a potent non-destructive tool for evaluating the physical and mechanical properties of conductive nano-reinforced cementitious nanocomposites.

## 1. Introduction

Cementitious composites have found extensive utility in numerous applications, primarily owing to their ready availability, cost effectiveness, straightforward production processes, and impressive compressive strength characteristics. Nevertheless, their inherent brittleness has posed a significant challenge, as cementitious materials tend to exhibit low tensile strength and fracture toughness. This brittle nature often leads to undesirable consequences, including cracking that can subsequently lead to structural and material degradation. In response to this challenge, substantial scientific efforts have been devoted to mitigating the brittle behavior of cementitious materials. Researchers have explored various concepts, capitalizing on the physical properties of both natural and synthetic macro- and microfibers [1]. These innovative approaches not only seek to enhance mechanical properties but also aim to fulfill the demand for multifunctional smart cementitious materials, offering improved crack-bridging, crack-healing, and sensing capabilities [1,2].

In conjunction with the recent strides in nanomaterial science, there has been a pronounced focus on investigating the impact of incorporating carbon-based nanomaterials (CBNs) into the cementitious matrix [3,4,5]. Owing to their distinctive characteristics, CBNs have gathered attention for their ability to enhance the microstructure of cementitious composites by facilitating the creation of a denser matrix. This enhancement arises from CBNs serving as nucleation sites for the deposition and growth of cement hydration products [4]. Consequently, micro-porosity and structural flaws are typically reduced, resulting in an overall improvement in mechanical performance. Furthermore, CBNs possess noteworthy electrical properties that can be harnessed to craft cementitious nanocomposites with apparent conductivity and piezo-resistive characteristics [2,4]. These tailored electrical attributes hold the potential to furnish valuable insights into the stress–strain conditions of the material or the early detection of potential damage.

Among the various CBNs, a widely used nanofiller in cementitious composites is graphene, a two-dimensional sheet of carbon atoms arranged in a hexagonal lattice, with Young’s modulus and mechanical strength as high as ~1 TPa and ~130 GPa, respectively [6]. Graphene and its derivatives can be produced in large quantities from graphite using top-down (i.e., detachment of the graphene layers from graphite) mechanochemical processes [7,8,9,10]. Graphene nanoplatelets (GnPs) are exfoliated from pristine graphite and consist of several layers of graphene sheets with diameters of several microns and thicknesses up to 100 nm. GnPs are known to contribute to improving the mechanical, thermal and electrical properties of various materials, including cementitious composites [11]. Moreover, GnPs have a reduced production cost compared with other CBNs, such as carbon nanotubes (CNTs) [12]. Hence, their excellent mechanical and electrical properties together with their availability and low cost renders them as an optimal option for incorporation in cementitious composites for improving the mechanical response and providing self-sensing properties.

The homogeneous dispersion of GnPs within the cementitious matrix emerges as a critical concern. The tendency of GnPs to agglomerate, driven by van der Waals interactions, poses a potential detriment to cementitious composites. Such agglomeration can introduce flaws and pores, serving as points for crack initiation [13] and subsequently exerting adverse effects on both the mechanical strength and electrical conductivity. Consequently, various experimental techniques detailed in the literature have been developed to assess the dispersion of GnPs, either prior to their incorporation [14,15,16,17] or subsequent to it [15,16,17,18,19,20,21,22], into the cementitious matrix. One notably effective method, frequently reported in the literature [13,14,15,23,24], involves subjecting graphene to ultrasonication energy in the presence of superplasticizers within a water medium. This procedure ensures the proper dispersion of graphene, mitigating agglomeration and enhancing the performance of cementitious composites.

To explore and understand the ways that the cementitious matrix is affected by the incorporation of GnPs, extensive laboratory studies including mechanical and electrical tests have also been performed on hardened cementitious nanocomposites. These studies are often conducted through compressive and/or flexural strength tests [18,19,20,21,25,26] and direct current (DC) resistivity measurements [27,28,29]. Mechanical testing reveals enhanced compressive and flexural strength up to a specific GnP concentration, which can significantly vary among the different studies. The improved performance was mostly attributed to the denser microstructure of the matrix resulting from the addition of GnPs and was also supported by porosimetry measurements [15,16,22,30]. On the contrary, above a certain nanofiller content, the mechanical response was decreased, presumably due to the excessive agglomeration of the nanoparticles. Correspondingly, the electrical resistivity of the cementitious nanocomposites was also affected in line with the GnPs content. In fact, electrical resistivity, in DC measurements, progressively decreased as the GnP content increased, until a critical volume fraction, above which it remained stable or even increased. This denotes that the continuous conductive network was interrupted by the formation of agglomerates and/or flaws in the cementitious matrix.

Electrical impedance spectroscopy (EIS) has found wide-ranging applications in the realm of cementitious composites, serving as a valuable tool for monitoring various facets of the material. These applications range from studying the cement hydration process [31,32,33] to tracking the evolution of microstructure [34,35] and observing degradation phenomena [36,37,38,39]. Nevertheless, in the context of evolving cementitious nanocomposites, EIS measurements have taken on an additional role. They are employed to investigate the dispersion state of carbon-based nanomaterials (CBNs), offering crucial insights into the percolation threshold [40,41]. Furthermore, EIS proves its worth by facilitating the monitoring of the hydration process and the detection of alterations in the microstructure [42]. Remarkably, it has emerged as a non-destructive method capable of evaluating the mechanical characteristics of cement composites [42,43].

The present research examines the impact of exfoliated GnPs (xGnPs) within the hardened cementitious matrix by EIS measurements alongside flexural tests, mercury intrusion porosimetry, and liquid nitrogen adsorption porosimetry analyses. Beyond the comprehensive examination of the physical, electrical, and mechanical features following xGnP incorporation, this study aims to elucidate a noteworthy correlation between EIS results and experimentally acquired parameters, specifically fracture toughness and porosity. An innovative perspective emerges as electrical measurements demonstrate their potential as a non-destructive tool for assessing these pivotal features of cementitious composites.

## 2. Materials and Methods

### 2.1. Preparation of Cementitious Nanocomposites

Cementitious nanocomposites were fabricated using type I ordinary Portland cement (CEM I 42.5 R) and graphene nanoplatelets of grade M (xGnPs) that had an average particle size of 5 μm and a thickness ranging from approximately 6 to 8 nm. These graphene nanoplatelets were sourced from XG Sciences Inc. (East Lansing, MI, USA) in Michigan. A superplasticizer (SP) based on polycarboxylate (ViscoCrete^®^ Ferro 1000, provided by Sika Hellas ABEE - Protomagias 15, 145 68, Kryoneri, Attica, Greece) was used for the current experimental program. The SP selection was based on the results obtained by previous research [23], as ViscoCrete^®^ Ferro 1000 proved to be the most suitable for uniform dispersion of xGnPs in the cementitious matrix among another three types of polycarboxylate-based superplasticizers provided by the same company.

To achieve a homogenous dispersion, the xGnPs and the SP were introduced into water, constituting 6% of the total mixing water volume. Ultrasonic energy was applied using a probe ultrasonicator to facilitate the dispersion of xGnPs within the aqueous solution. After achieving the desired dispersion of xGnPs, additional water was added to attain a specified water-to-cement ratio (w/c) of 0.3. The resulting mixture was manually stirred and combined with the cement, following the procedures outlined in ASTM C305 [44]. Five (5) distinct formulations were prepared, each containing varying xGnP contents: 0.0%, 0.05%, 0.10%, 0.20%, and 0.40%, based on the dry weight of the cement and named M-0.00, M-0.05, M-0.10, M-0.20, and M-0.40 respectively; the number after “M-” identifies the xGnP percentage concentration (mixture). The detailed mixture compositions are presented in Table 1.

As depicted in Figure 1, two types of prismatic specimens, each measuring 20 × 20 × 80 mm, were manufactured for the purpose of investigating (a) the mechanical and (b) the electrical characteristics of the prepared mixtures, respectively. Notched prismatic specimens were utilized for conducting mechanical tests, employing a three-point bending setup. Meanwhile, prismatic specimens were employed for the electrical resistance tests and were equipped with two embedded steel mesh electrodes. These steel mesh electrodes were positioned at an approximate distance of 30 mm from each other. All specimens were released from their moulds 24 h post-casting and were subsequently submerged in a saturated Ca(OH)_2_ solution (0.82 g/L) to prevent the leaching of calcium ions from the specimens. This immersion process was maintained at room temperature (20 ± 2 °C) until the specimens reached the age of 28 days.

### 2.2. Experimental Setup

#### 2.2.1. Electrochemical Impedance Spectroscopy Measurements

EIS and equivalent circuits were employed to examine the conductive mechanisms within cementitious matrices reinforced with xGnPs. These analyses were conducted on prismatic specimens containing two embedded steel mesh electrodes, as illustrated in Figure 1b. To ensure nearly complete saturation, all specimens were immersed in tap water for a duration of 7 days under ambient atmospheric conditions prior to conducting the measurements. EIS was carried out using the Dielectric Thermal Analysis System (DETA-SCOPE^®^, ADVISE), where an alternating voltage signal with an amplitude of 1 V was applied. The frequency of the signal was swept in the range of 1 Hz to 0.391 MHz. The magnitude of impedance and the phase angle were recorded using the data acquisition software of the system. Utilizing the data obtained from these scans, Nyquist plots were generated. These plots depicted the negative imaginary component against the real part of impedance, spanning the corresponding frequency range.

The obtained spectra were analysed using equivalent circuit (EC) models with EIS Spectrum Analyser software [45]. First proposed in the late 1980s, this is a common mechanistic approach to study the electrical characteristics of cementitious materials [46].

#### 2.2.2. Mechanical Tests

Flexural (three-point bending) experiments were carried out on notched prismatic specimens, as depicted in Figure 2. At least three (3) specimens were investigated to obtain a reliable average value of the mechanical behaviour. Following mechanical testing, the fracture toughness (*K*_Ic_) of each specimen was determined. The bending tests were conducted in the first place for material characterization, to investigate the effect of the inclusion of the xGnPs in the cement matrix. Secondly, the calculated values could also serve as indicators of the quality of the xGnP dispersion in the cement paste and might also reveal a correlation between the mechanical and electrical responses of the cementitious nanocomposites.

The tests were carried out using an MTS (MTS Systems Co., Eden Prairie, MN, USA) testing machine with a 10 kN capacity. During the testing process, the mechanical load was applied at the three points of the specimen, as indicated in Figure 2, with a constant crosshead displacement rate of 0.0015 mm/s. Simultaneously, measurements of applied load, vertical displacement, and crack mouth opening (CMOD) were recorded.

The *K*_Ic_ is given by Equation (1) and calculated by using Equation (2) according to ASTM E399 [47], as:*K*_Ic_ = *P*_max_∙*S*∙*f*(*a/W*)/(*B∙W*^1.5^)(1)
*f*(*a/W*) = 3(*a/W*)^0.5^∙(1.99 − (*a/W*)∙(1 − *a/W*))∙(2.15 − 3.93(*a/W*) + 2.7(*a/W*)^2^)∙(2 + 4∙*a/W*)∙(1 − *a/W*)^−1.5^(2)
where *P*_max_ is the ultimate load, *S* is the span of the specimen, *B* is the width of the specimen, *W* is the height of the specimen, and *α* is the depth of the notch.

#### 2.2.3. Porosity Measurements

After the completion of flexural test series, the broken parts of each mixture were picked up randomly to measure the porosity and pore size distribution using nitrogen adsorption and mercury porosimetry.

##### Nitrogen Adsorption Study

Investigations were conducted using N_2_ adsorption isotherms at 77 K to ascertain the structural attributes of the samples, including pore size distribution, total pore volume, and specific surface area. The experiments were conducted using a Quantachrome Autosorb IQ instrument. Before performing the isotherm measurements, the samples underwent overnight outgassing at 120 °C.

##### Mercury Porosimetry

Mercury porosimetry measurements were performed on specimens from vacuum up to pressure of 60,000 psig. The experiments were carried out using a PoreMaster 33 instrument from Quantachrome Instruments, Boynton Beach, FL, USA, as a standard procedure for analyzing the pore structure of solids encompassing pore sizes ranging from large mesopores to macropores. The technique is precisely described by Gregg and Sing [48]. The equivalent pore radius r was computed according to the capillary pressure Washburn equation [49]:*R* = −3∙(2*γcosθ*)/*P_C_*(3)
where *P_C_* is the capillary pressure in Pa, *γ* is the interfacial tension in Nm^−1^ (*γ*_Hg/air_ = 0.471 Nm^−1^), and *θ* is the contact angle (*θ_γ_*_Hg/air_ = 140°, as suggested by Cook and Hover [50]). The useful information obtained from the mercury porosimetry intrusion curves mainly concerned the total pore volume of the sample and the pore size distribution.

## 3. Results and Discussion

### 3.1. Electrochemical Impedance Spectroscopy

#### 3.1.1. Nyquist Plots and Electrical Resistivity

The EIS experimental findings for cementitious composites with different xGnP concentrations can be illustrated through Nyquist plots, as displayed in Figure 3, where the real and the negative imaginary part of the impedance are presented. The impedance, which expresses the resistance of the AC to flow within the cementitious matrix, is determined as follows:*Z* = *Z′* + *Z″j*,(4)
where *Z* is the complex impedance, *Z′* is the real part (ohmic resistance), *Z″* is the imaginary part (reactance) of impedance, and *j* = $\sqrt{-1}$. *Z′* and *Z″* have been calculated using Equations (5) and (6), respectively:*Z′* = |*Z*| ∙ *cos*(*arc*{*Z*}*∙π*/180)(5)
*Z″* = −|*Z*| ∙ *sin*(*arc*{*Z*}*∙π*/180)(6)

In each plot, the test frequency was scanned by decreasing from 0.391 MHz to 1 Hz, with corresponding data points spreading from left to right with respect to the horizontal axis. As can be seen in Figure 3, every Nyquist plot comprises of two arcs, with the right-most (lower-frequency) arc corresponding to the geometrical capacitance formed by the two steel electrodes and the second (higher-frequency) arc being associated with the formed electrochemical double layers at the electrolyte/electrode interfaces and the electrolyte ohmic resistance in the pores of the impregnated cementitious composites. Moreover, the point where the two arcs meet (cusp point) and where the reactance, representing the energy storage (capacitive) part [41] of the material, is almost zero, can be described as the total ohmic resistance of the cell, which is the sum of the electrolyte’s ohmic resistance *R*_s_ and the charge transfer resistance *R*_ct_ [46]. Based on the Nyquist plots depicted in Figure 3, it is evident that the reference mixture displays the highest total ohmic resistance, whereas the mixtures enhanced with xGnPs exhibit a noticeable decrease in resistance. It is also noticeable that the cusp points for the mixtures M-0.05 and M-0.10 tend to show comparable values of approximately 2.5 kOhm, whereas the same applies for the other two mixtures (M-0.20 and M-0.40), showing higher total resistance values of nearly 4.0 kOhm.

The electrical resistivity (*ρ*_AC_) for each specimen was calculated by using the following equation:*ρ*_AC_ = *R∙A/l*,(7)
where *R* is the ohmic resistance at the cusp point, *A* is the cross section of the specimen, and *l* is the distance between the internal electrodes.

As can be derived from Figure 4, the three-sample average total electrical resistivity values, obtained from the corresponding cusp points of the Nyquist plots of the investigated mixtures, first decrease with xGnP content and then increase up to an intermediate constant value.

#### 3.1.2. Equivalent Circuit Analysis and Results Discussion

Several EC models have been introduced in the existing literature to effectively match Nyquist plots and interpret the impedance response observed in cementitious composites reinforced with carbon-based nanomaterials (CBNs) [22,29,40,42,43]. Figure 5a presents a Randles circuit, which is often used to describe electrochemical systems in different research fields.

In this EC model, the series resistance, denoted as *R*_S_, signifies the ohmic resistance within the electrolyte solution, *R*_ct_ represents the charge transfer resistance, and *Q* corresponds to the constant phase element’s admittance *Y* = 1/|*Z*| at ω = 1 rad/s in F∙s^−1^. *Q* is intricately connected to the double layer capacitance at the interface between the electrode and the liquid electrolyte, addressing their interactions and characteristics. *W* stands for the Warburg resistance caused by ion diffusion in the electrolyte. Nevertheless, this simple model was not appropriate to describe the electrochemical response of the nano-reinforced specimens of the current research; therefore, the modified Randles circuit of Figure 5b was used instead, which includes three additional elements in a second loop. The second EC model considers the overall specimen’s geometrical capacitance in terms of an xGnP-related capacitor *C*_N_ in parallel connection with a cementitious matrix-related non-ideal capacitor comprising of *R*_int_ and *C*_int_ elements in series connection.

As mentioned previously, *R*_S_ is used to indicate the resistance of the electrolyte (water) that is present in the pores of the cementitious matrix, as shown in Figure 6. *R*_s_ mainly depends on three parameters:(i)The volume of electrolyte-filled pores in the bulk material far from the electrodes. The higher the pore volume, the easier the passage of ions from electrode to electrode and the smaller the series (electrolyte) resistance [29].(ii)The concentration of dissolved ions in the electrolyte, which might be enhanced by oxygen groups at the filler surface; this can be ignored here since the xGnPs used had no such groups.(iii)Partial percolation of conductive filler xGnPs, forming electron pathways that extend from the electrodes’ surfaces and shorten the distance between them, causing *R*_s_ to drop. This becomes the case when agglomerates form, as is illustrated in a forthcoming figure.

Figure 7 shows the average values of electrolyte resistance for the different mixtures. Interestingly, the pore solution resistance reveals a tendency to decrease as the xGnP concentration increases, which could be partially associated with the increasing pore volume at lower xGnP concentrations and partially with agglomerate formation causing electrode extension and shorter pathways for ions (as will be shown later in this section) at higher xGnP concentrations.

On the other hand, the charge transfer resistance *R*_ct_, plotted versus xGnP concentration in Figure 8, is mainly a function of the amount of charge that transfers at the electrode/electrolyte interface. As follows, it is a function of the active surface area of the electrode, i.e., the electrode’s surface area that is in contact with electrolyte. Obviously, the higher the active surface area of the electrode, the smaller the charge transfer resistance. As can be derived from the following sections, with the increase in xGnP content, initially, both microporosity and mesoporosity increase. This increase manifests itself also as an increase in the active electrode surface area and therefore perfectly justifies the initial reduction in *R*_ct_ up to a 0.10% xGnP concentration. Nevertheless, the increase in *R*_ct_ as xGnP concentration further increases from 0.10% to 0.40% can only be explained as the result of formation of hydrophobic xGnP agglomerates, which obstruct electrolyte access to parts of the electrodes’ surface, as shown in Figure 9b. It is notable that although an increase in the highly microporous xGnP concentration leads to micropore volume enhancement (Table 2), their hydrophobic nature (oxygen < 1%) practically blocks the passage of water through their slit-like microporous structure and, at the same time, their adhesion with the hydrophilic concrete matrix is not expected to be tight, leaving gaps at the matrix/filler interface. These gaps are estimated to be in the range of mesopores, which fill with electrolyte and, in agreement with mercury porosimetry results (Table 2), become less in terms of mesopore volume upon agglomerate formation, as shown in Figure 9a.

### 3.2. Porosity Studies

In the current research, porosimetry tests were conducted to examine the effect of the addition of specific concentrations of xGnPs in the total porosity and pore size distribution of the cementitious matrix. According to the literature, CBNs act as nucleation points where C-S-H particles can grow. As a result, the cementitious microstructure is often denser, exhibiting reduced total porosity. In case that there is an excessive clustering of the nanoparticles, the total porosity of the nanocomposites can increase, and the pore size distribution can alter due to pores created by inefficient nanoparticle distribution.

Overall, mercury adsorption porosimetry provided the pore volume and surface area of mesopores, whereas nitrogen adsorption isotherms at 77 K provided the respective parameters for micropores. The porosimetry parameter results versus xGnP content are summed up in Table 2. In general, the amounts of adsorption were low (Figure 10), with LN_2_-porosimetry-calculated BET surface areas (Brunauer–Emmett–Teller) ranging between 7.6 and 14 m^2^/g and the measured total adsorbed N_2_ amounts fluctuating between 33 and 56 cc/g (STP).

In the analysis of pore size distribution, the BJH model [51] was employed, focusing on the desorption branch of the LN_2_ isotherms. The determined pore sizes (diameters) fell within the range 2 to 100 nm for all five examined samples. These samples exhibited similar isotherms, characterized as type IV with an H3 hysteresis loop, as depicted in Figure 10. The main difference among the five isotherms was the different total absorbed amounts and, in the case of sample “M-0.20”, which has the highest BET area and largest pore volume, also a slightly but evidently wider hysteresis loop. The pores that are added with the increasing xGnP content have pore diameters mainly in the range 4–100 nm. Additionally, the existence of a reproducible, permanent hysteresis loop in all cases is generally associated with capillary condensation and, in this present case, can be attributed to network effects [52]. The type of the presented hysteresis loop H3 corresponds to non-rigid aggregates of plate-like particles (e.g., certain clays) but also to macropores that are not completely filled with pore condensate. In the studied samples, both of the abovementioned cases can co-exist.

As can be derived by comparing the pore volumes given in Table 2, in all samples the main part of the total pore volume is micropores. Nevertheless, upon a xGnP content increase, the micropore volume was also found to increase, whereas the mesopore volume initially increased up to a 0.10 wt.% xGnP content and then dropped to an intermediate value; this was attributed to agglomerate formation at the higher concentrations, as described in Section 3.1.2 (Figure 8).

### 3.3. Mechanical Properties

Figure 11 shows the average fracture toughness of 28-day-old cement specimens. Overall, the fracture toughness shows elevated average values for the two mixtures reinforced with xGnPs compared with the plain mixture. The increase demonstrates a peak value for the M-0.05 (29% increase); then, for the higher xGnP contents (from 0.10 wt.% to 0.40 wt.%), the average *K*_Ic_ values start to drop, without reaching the reference mixture corresponding value.

The mechanical behaviour observed in the present study, showing an initial enhancement for mixtures with the lowest xGnP concentrations followed by a gradual decline as the nanoparticle content increases, aligns with findings reported in prior studies involving comparable GnP concentrations [16,19,30]. The enhanced toughness observed in the mixtures M-0.05 and M-0.10 can be attributed to the effective inhibition of micro-crack propagation, which is a result of the well-dispersed nanoparticles within the cementitious matrix. Conversely, less efficient dispersion, which may occur at higher nanoparticle concentrations, can compromise the flexural response of the cementitious nanocomposites. In fact, inadequate dispersion of xGnPs may lead to the formation of agglomerations, thereby contributing to an inferior mechanical performance.

### 3.4. Correlationship between Electrical and Mechanical Properties

Furthermore, from the comparison of the graphical curves of *ρ*_AC_ and *K*_Ic_ as a function of xGnP concentration (Figure 4 and Figure 11), it was observed that the two curves have an inverse response, which was revealed by the linear plot of *K*_Ic_ versus 1/*ρ*_AC_ (Figure 12) regarding the corresponding xGnP concentrations. The equation that describes the functional relationship between 1/*ρ*_AC_ and *K*_Ic_ is given in Equation (8).
*K*_Ic_ = 0.016/*ρ*_AC_ + 0.7237(8)

Nevertheless, the aforementioned equation represents distinct material and experimental factors, including variations in the nanofiller type and concentration, the type of cement and superplasticizer employed, the water-to-cement (w/c) ratio, the specimen dimensions, and various other material and experimental attributes. This indicates that both the 1/*ρ*_AC_-coefficient (0.016) and the constant value (0.7237) within the regression equation are anticipated to undergo modification when different nanomaterials and experimental conditions are chosen.

Recognizing that the correlation established in the present research article is contingent upon the specific material and experimental parameters employed, it is evident that further research is warranted to assess the potential transferability of this methodology to various types of cements and carbon-based nanofillers (CBNs). This avenue of inquiry holds promise for broadening the scope of fracture toughness assessment in cementitious nanocomposites. Nonetheless, it is evident that EIS can offer a straightforward approach that can be readily utilized for the assessment of fracture toughness in conductive cementitious nanocomposites.

## 4. Conclusions

In the present research article, electrochemical impedance spectroscopy (EIS) measurements were employed to examine multiple properties of graphene-based cementitious nanocomposites, including resistivity, porosity, and fracture toughness. During our experimental research, several noteworthy conclusions were drawn:-The electrical resistivity of the investigated cement mixtures exhibited a distinctive pattern. Initially, as the content of xGnPs increased (ranging from 0.05 wt.% to 0.10 wt.% by cement), the resistivity decreased. Nevertheless, for mixtures incorporating up to 0.40 wt.% xGnPs, a subsequent increase occurred, stabilizing at an intermediate constant value when compared with the reference mixture.-There was a visible trend in the reduction of pore solution resistance (*R*_s_) as the xGnP concentration increased. This phenomenon can be attributed, in part, to the expansion of pore volume at lower xGnP concentrations and the formation of agglomerates. These agglomerates extended the electrode interface and consequently led to shorter ion pathways. Porosity measurements further substantiated the increase in total pore volume following the incorporation of xGnPs into the cement mixture.-Both EIS and porosimetry analyses provided compelling evidence of agglomerate formation, particularly when xGnP concentrations exceeded 0.10% by dry cement weight.-A functional linear relationship was observed between fracture toughness, as assessed through bending tests, and electrical resistivity, determined via EIS measurements.

Consequently, this study establishes the applicability of EIS as a valuable non-destructive tool for the comprehensive assessment of the physical and mechanical properties of cementitious nanocomposites.

## Figures and Tables

**Figure 1 nanomaterials-13-02652-f001:**
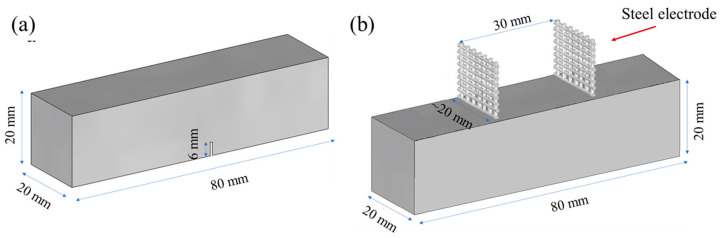
Prismatic specimens used for (**a**) mechanical tests (three-point-bending) and (**b**) electrochemical impedance spectroscopy measurements.

**Figure 2 nanomaterials-13-02652-f002:**
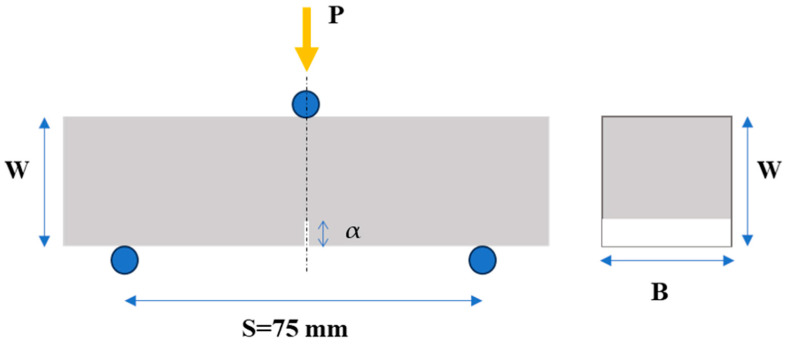
Flexural (three-point bending) test set-up and specimen key dimensions (W: height, B: width, S: span, and α: depth of the notch of the specimen) used for calculation of fracture toughness (*K*_Ic_).

**Figure 3 nanomaterials-13-02652-f003:**
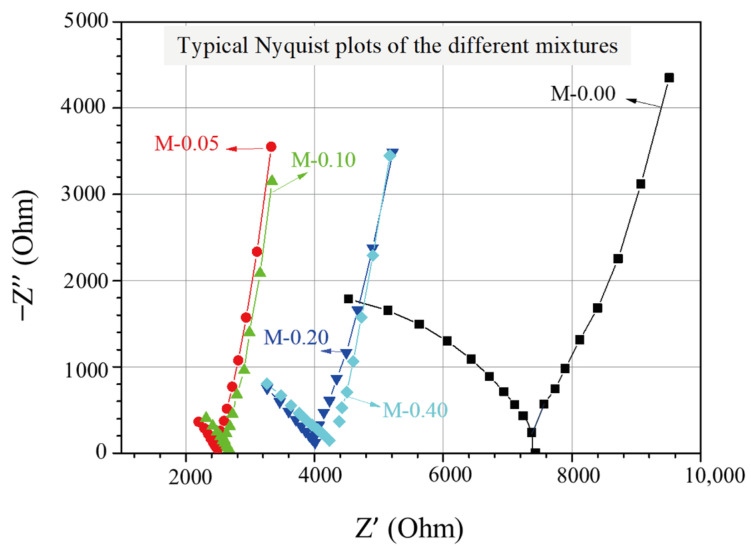
Typical Nyquist plots of the investigated nanoreinforced cement specimens.

**Figure 4 nanomaterials-13-02652-f004:**
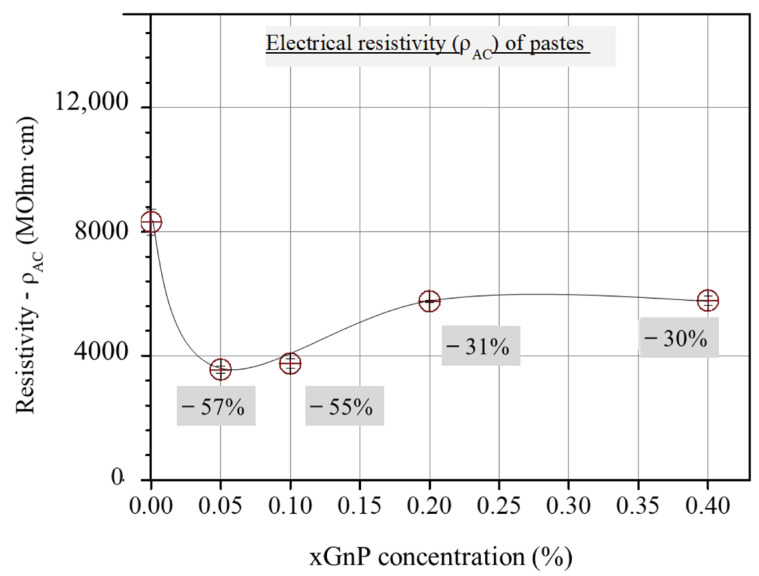
Τotal ohmic resistivity from Nyquist cusp points vs. xGnP concentration.

**Figure 5 nanomaterials-13-02652-f005:**
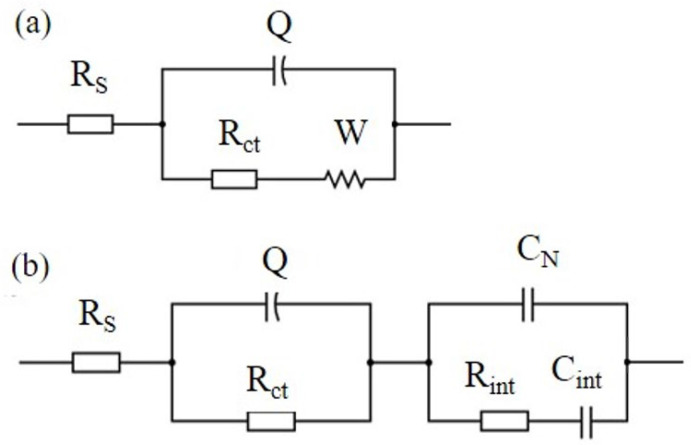
(**a**) EC model (Randles circuit) describing cement paste without xGnPs and (**b**) EC model describing cement pastes with xGnPs.

**Figure 6 nanomaterials-13-02652-f006:**
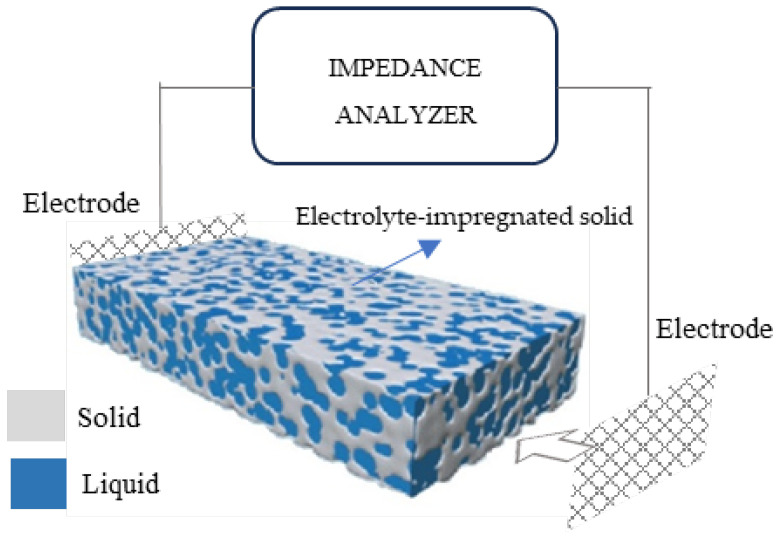
Set-up representation for EIS measurements on porous electrolyte-impregnated solid.

**Figure 7 nanomaterials-13-02652-f007:**
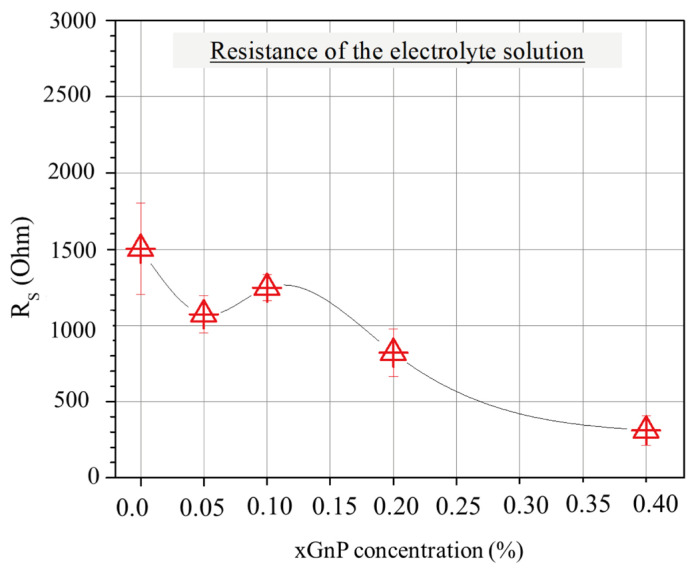
Average resistance of electrolyte solution according to EIS for mixtures with various xGnP concentrations, as obtained by fitting Nyquist plots with the EC model of Figure 4b.

**Figure 8 nanomaterials-13-02652-f008:**
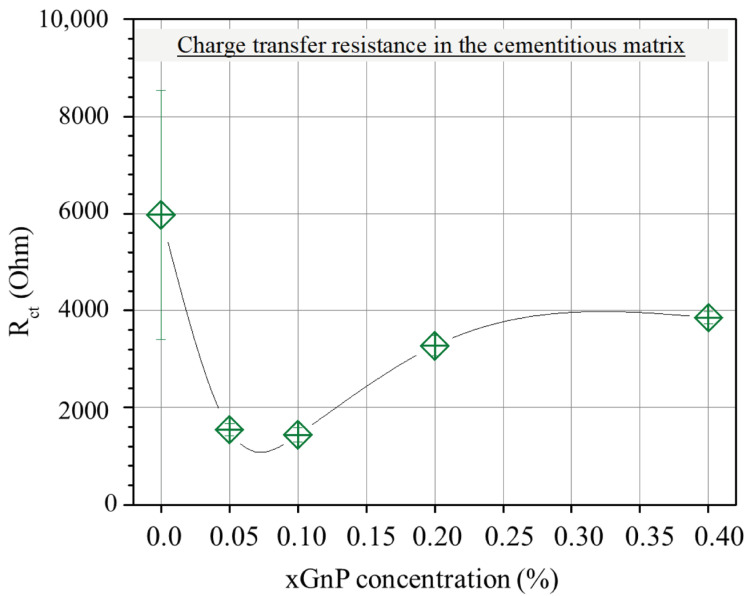
Average resistance (*R*_ct_) in the cementitious matrix with varying xGnP concentration, as obtained by the EC model.

**Figure 9 nanomaterials-13-02652-f009:**
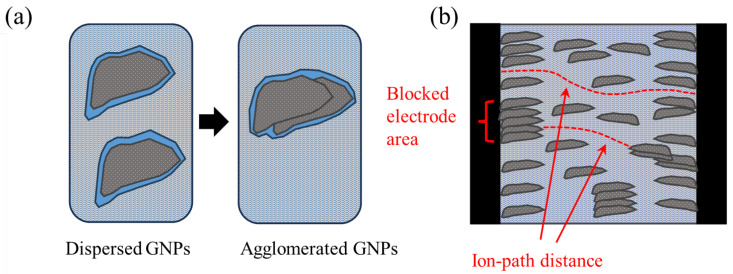
Schematics representing (**a**) mesoporosity reduction upon agglomeration and (**b**) agglomerate-induced electrode active area reduction, as well as ion path distance reduction. The blue color represents electrolyte (tap water), light grey represents concrete, dark grey represents xGnP filler, and black represents steel (electrodes).

**Figure 10 nanomaterials-13-02652-f010:**
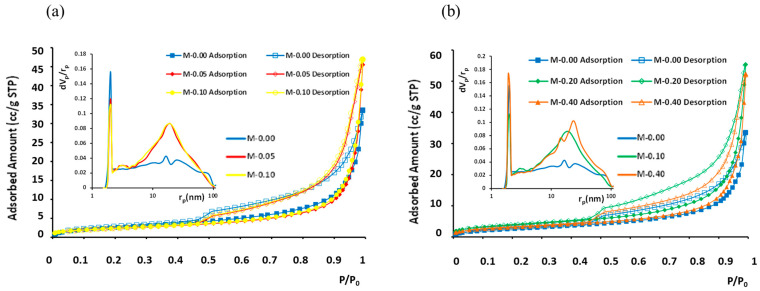
N_2_ adsorption isotherms at 77 K and BJH pore size distribution curves (inset) for different xGnP concentrations: (**a**) 0–0.1 wt.% and (**b**) 0–0.40 wt.%. Filled symbols stand for adsorption branch and empty symbols for desorption branch.

**Figure 11 nanomaterials-13-02652-f011:**
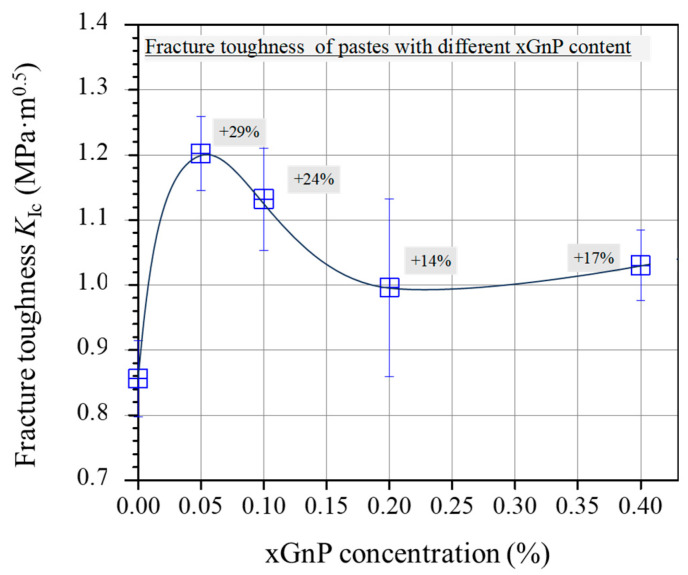
Average fracture toughness (*K*_Ic_) of pastes with different xGnP concentrations.

**Figure 12 nanomaterials-13-02652-f012:**
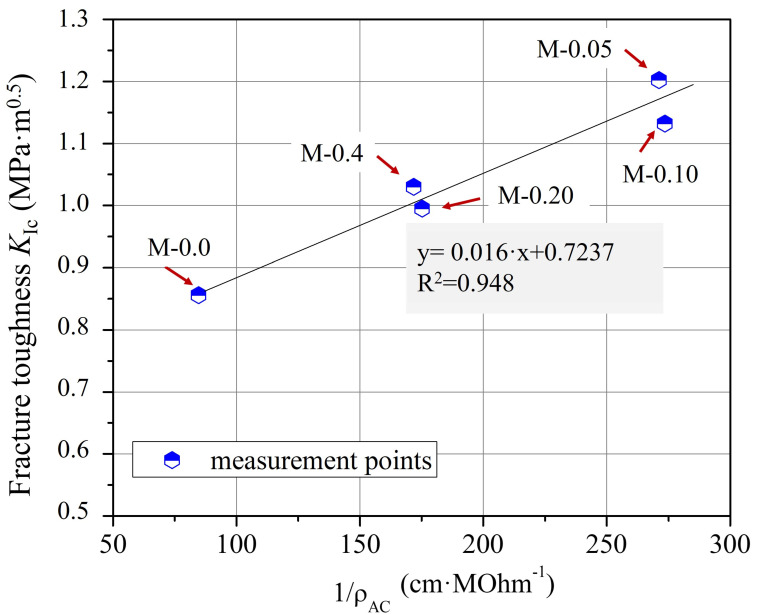
Correlation between *K*_Ic_ and 1/*ρ*_AC_ and the fitting curve for cementitious composites with varying xGnP concentrations.

**Table 1 nanomaterials-13-02652-t001:** Mixture compositions used in the present investigation.

	Materials			
Mixture	CEM I	Water	xGnPs	SP
	(g)	(g)	(g)	(g)
M-0.00	380	114	0.0	-
M-0.05	380	114	0.19	1.52
M-0.10	380	114	0.38	3.04
M-0.20	380	114	0.76	6.08
M-0.40	380	114	1.52	13.16

**Table 2 nanomaterials-13-02652-t002:** Exportable data from mercury and nitrogen porosimetry.

Mixture	Mercury Porosimetry						N_2_ Porosimetry		
	ε (%)	Density (g/cm^3^)		Pore Volume (cm^3^/g)	s.s.a ^1^ (m^2^/g)	Pore Radius (nm)	Pore Volume (cm^3^/g)	Pore Radius (nm)	s.s.a (BET) (m^2^/g)
		Bulk	Apparent						
M-0.00	4	2.14	2.23	0.0187	6.078	4–3665	33.50	4–200	10.08
M-0.05	18	2.13	2.60	0.0846	34.68	4–2655	45.50	4–200	7.60
M-0.10	18	2.09	2.54	0.0865	39.18	4–3270	46.95	4–200	8.85
M-0.20	12	2.12	2.44	0.0606	27.98	4–2730	55.35	4–200	14.04
M-0.40	16	2.09	2.50	0.0771	30.46	4–2690	52.30	4–200	11.4

^1^ Specific surface area.

## Data Availability

Data are available upon request to the authors.

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
