# Peer review of "Electrochemical Impedance as an Assessment Tool for the Investigation of the Physical and Mechanical Properties of Graphene-Based Cementitious Nanocomposites"

_nanomaterials, 2023, doi:10.3390/nano13192652_

Round 1

Reviewer 1 Report

The paper titled “Electrochemical impedance as an assessment tool for the investigation of the physical and mechanical properties of graphene-based cementitious nanocomposites” reports the impact of graphene nanoplatelets within the hardened cementitious matrix by Electrochemical Impedance Spectroscopy measurements alongside flexural tests, mercury intrusion porosimetry and liquid nitrogen adsorption porosimetry analyses as well as a correlation between EIS results and such parameters as fracture toughness and porosity. The research is excellent from experimental point of view, and the manuscript is well written. Therefore, I recommend it for publication after minor revision. The comments are listed below.

1.     The main question that remained unclear is how general is the revealed correlation. Can these results be applied for some other types of cements and CBNs? Or, in each case, similar research should be performed. The prospective for the application of the proposed method should be clarified.

2.     Technical remarks:

Line 142: “g/l” -> “g/L”

The subsection numbering is erroneous. For instance, sections 2.2.2. and 3.2 are missing.

Line 180: “shown in Figure , with” -> “shown in Figure 2, with”

Line 185-186: “where Pmax is the ultimate load, S the span of the specimen, B the width of the specimen, W the height of the specimen, and α the depth of the notch.” -> “where Pmax is the ultimate load, S is the span of the specimen, B is the width of the specimen, W is the height of the specimen, and α is the depth of the notch.”

Line 203: “γ the interfacial tension in” -> “γ is the interfacial tension in”

Line 214: “Z' the real part (ohmic resistance) and Z'' the” -> “Z' is the real part (ohmic resistance) and Z'' is the”

Ref. 42 seems to be incorrect. “International, A.” could not be surname and name of the author.

Refs. 20 and 41 are duplicates.

Reviewer 2 Report

The paper reports the conclusion of a study of the effect on physical and mechanical properties (such as resistivity, porosity, and fracture toughness) of the inclusion of graphene platelets into a concrete matrix, performed through electrochemical impedance spectroscopy, mechanical tests, and porosity measurements. In particular, the relevance of electrochemical impedance spectroscopy for this kind of non-destructive analyses is highlighted.

The paper is well-written and useful both from a theoretical and a practical point of view.

Therefore, I suggest its publication with some minor corrections and modifications that I detail in the following.

1) line 20: "indicated" -> "indicated by"

2) lines 22 and 419: "resistivity" -> "conductivity" (the inverse of resistivity)

3) line 56: a few references on different applications and studies of graphene could be useful for the interested reader; I suggest to cite:

Novoselov, K.; Fal'ko, V.; Colombo, L.; Gellert, P. R.; Schwab, M. G.; Kim, K. A roadmap for graphene. Nature 2012, 490, 192–200. DOI: 10.1038/nature11458

Marconcini P.; Macucci M. Transport Simulation of Graphene Devices with a Generic Potential in the Presence of an Orthogonal Magnetic Field. Nanomaterials 2022, 12, 1087. DOI: 10.3390/nano12071087

Huang, X.; Yin, Z.; Wu, S.; Qi, X.; He, Q.; Zhang, Q.; Yan, Q.; Boey, F.; Zhang, H. Graphene-based materials: synthesis, characterization, properties, and applications. Small 2011, 7, 1876-1902. DOI: 10.1002/smll.201002009

4) line 79: "enhancement of" -> "enhancing"

5) line 81: "Exploring and understanding the ways that" -> "In order to explore and understand the ways in which"

6) line 108: when using "xGnP" for the first time, it would be useful to clarify that "x" is for "exfoliated"

7) line 182: "alpha" appears in a wrong position

8) equations (1) and (2): in these equations "alpha" is not well identifiable; is it possible to solve this problem, for example using a different font?

9) equation (4): the Authors should clarify the meaning of "i" and "j" (the unit vectors along the two axes of the Gauss plane?)

10) lines 230-232: the Authors should specify that the number after "M-" identifies the xGnP percentage concentration

11) figure 5(b): the segment identifying the right contact is missing and should be added

12) line 256: I read "in F s^(n-1)"; probably it should be "in F s^(-1)"

13) lines 320 and 380: the number of these sections should be changed to "3.2" and "3.3", respectively

14) line 366: "until and" -> "to" (maybe)

15) line 384: "regarding the" -> "for" (maybe)

16) line 393: should "either...or" be substituted by "both...and" (I imagine both the coefficients should be changed)

17) line 397: in the Authors' opinion, could these conclusions (for example the proportionality of Equation (8)) be valid also for different conductive nanofillers?

Reviewer 3 Report

Graphene-based cemented nanocomposites have attracted much attention due to their physical and structural properties, include resistivity, porosity and fracture toughness. The related properties of cement mixtures with different concentrations of graphene nanosheets can be effectively evaluated by electrochemical impedance spectroscopy. In this paper, there are the following problems were suggested to modify.

1. The description in Figure 2 is not detailed enough, so it is suggested to supplement it.

2. When comparing the samples in Figure 3, it is recommended to convert them into data tables, so that the comparison effect will be better.

3. According to Figure 7, it is concluded that the resistance of pore solution shows a downward trend. Why does it rise between 0.05 and 0.1?

4. The scale format of the X-axis in Figure 7 and Figure 8 is wrong, so it is suggested to modify it.

5. Some excellent works such as Energy (2023, 270: 126880), Energy Mater. (2021, 1: 100006), and Energy Mater. (2023, 3: 300016) should be cited.

The full text should be checked. 

Round 2

Reviewer 3 Report

In the case of the revised ms, it was improved. However, in the case of the reply to comment 5, it is not correct since it is related to accessment. As a result, simulation and materials are very important. It is recommended to cite these works so that more readers can show great interest to this work. 

Some excellent works such as Energy (2023, 270: 126880), Energy Mater. (2021, 1: 100006), and Energy Mater. (2023, 3: 300016) should be cited.